# MODELING KNOWLEDGE AS FUNCTIONALS FOR KNOWLEDGE REASONING

## ABSTRACT

A bottleneck for developing general artificial intelligence is empowering machines with knowledge-reasoning capabilities to facilitate NLP tasks such as semantic search, reading comprehension, and question-answering. Prior arts focus on integrating distributed knowledge embeddings and representations of pre-trained neural language models to produce outputs; however, there are still large areas for improvement in performance and sustainability. In this paper, we propose to represent **K**nowledge **as** the **F**unctional representation ($KasF$) with a dynamics-based mechanism that simulates the semantic flow amongst tokens to facilitate knowledge reasoning. The method utilizes a superposition of semantic fields to represent knowledge, by building a dynamical mechanism to compute the similarity between semantic units. This mechanism comprehensively captures the semantic features and eliminates ambiguities in representing entities and relations. We first evaluate our $KasF$ on the WikiQA dataset to demonstrate its superiority in capturing semantic patterns. Next, we evaluate our $KasF$ modules on the SQuAD2.0 dataset by replacing the last layer of pre-trained language models fine-tuned on this dataset. We observe consistent improvements in accuracy with fewer parameters. Then we evaluate $KasF$ on the CommonsenseQA benchmark[1]. On the official blind test set, we achieve state-of-the-art with a single model, outperforming the prior best ensemble and single models by $0.4\%$ and $3.1\%$, respectively[2]. It is worth noting that the prior best single model is $47\times$ larger than ours. Further experiments also demonstrate that $KasF$ exhibits superiority in dealing with sophisticated sentences.

## 1 INTRODUCTION

The ability of reasoning, especially knowledge reasoning, is essential for humans to think and interact with the world (Chen et al., 2020). The theory and application of knowledge reasoning have been studied for a long time (Zhang & Zhang, 1992; Kompridis, 2000), which claims that reasoning is the cognitive procedure of concluding existing facts by rules. Many successful cases (Silver et al., 2017; Bubeck et al., 2023) show that artificial intelligence models require robust and reliable reasoning capabilities. Thus, empowering machines with the abilities of knowledge reasoning can be regarded as imitating the ability of humans to use known knowledge, *i.e.*, factual statements retrieved from a knowledge base, to infer new knowledge and make decisions (Aamodt, 1990).

A dominant building block for knowledge reasoning is the large-scale language models (LLMs) that are powerful in natural language understanding (Min et al., 2021; Huang & Chang, 2023). Nevertheless, LLMs need an explicit mechanism to deal with knowledge-intensive information. On the other hand, knowledge graphs (KGs) succeed in using topological features among entities (Chen et al., 2020) to interpret knowledge-centric data. KGs are indispensable in providing context out of LLMs in the form of entities linked with substantial relations to produce the outputs. Mainstream knowledge reasoning approaches are methods coupling LLMs with KGs (Pan et al., 2023), includ-

---

[1] We only compare with the models that do not use ConceptNet since the CommonsenseQA's team officially no longer accepts this submission type. As ConceptNet was used to create the dataset, it filters the human-generated distractors, reducing the 5-way multi-choice to a simpler one.

[2] We will include the link to the latest official leaderboard after the anonymity period. Note that the leaderboard directly accessed from the website is outdated and has not been updated since 2021.

ing KG-BERT (Yao et al., 2019), KagNet (Lin et al., 2019), QA-GNN (Yasunaga et al., 2021), GreaseLM (Zhang et al., 2022), LLM-based KGs (AlKhamissi et al., 2022), and KG-augmented LLMs (Yang et al., 2023). They achieve improved accuracy by combining the merits of natural language understanding and structural knowledge guidance. However, there are still areas for improvement in performance and sustainability of knowledge reasoning.

In general, existing methods adopt the "knowledge as embeddings" strategy that trains the deep neural network with facts organized as triplets. The semantic units, *i.e.*, words/entities, sentences, or paragraphs, are represented as distributed vectors with fixed dimensions (Bengio et al., 2000; Mikolov et al., 2013; Devlin et al., 2018). Despite its computational efficiency, this strategy is quite different with the mechanism of the human brain that works as dynamic functions to represent semantic relations of a field of concepts (Patterson et al., 2007). In linguistics, this mechanism is interpreted as the *semantic field*, referring to a semantically structured group of the lexical set of words (Jackson & Amvela, 2007). Every token in a factual statement is equipped with varying semantic fields that express its relations to the other tokens. Inspired by the above biological and linguistic study, we model knowledge via functional representation. A functional is a real-valued function on a space of functions, *i.e.*, it takes functions as arguments. A functional can map linear mappings from a vector space into its field of vectors or scalars (Lang, 2012). Each semantic unit is treated as a functional that takes the objectives, the external states, and the internal states as the input arguments, and returns a task-specific encoding with varying structure/dimensions and roles.

Given the functional representation of semantic units, another bottleneck problem is enhancing entity connectivity and knowledge throughput for knowledge reasoning. Motivated by Pei & Wang (2023) that interprets neural weights as the path integrals between neuronal dynamics, we deal with the semantic relation between arbitrary semantic units as a dependent variable between them rather than an independent trainable parameter. As a result, our method can capture the necessary semantic patterns as much as possible, needless of storing every possible relation pair amongst the semantic units. To this end, we propose "knowledge as functions" (*KasF*), a dynamics-inspired mechanism, to simulate the dynamics amongst semantic units in human brain. Instead of edge-based entity relation, *KasF* links the knowledge-intensive entities via multiple weak but more flexible relations, preserving richer semantic inter-dependency. Rather than traditional knowledge retrieval by similarity ranking, *KasF* computes nonlinear trainable metric functions as the similarity between the functional representations of semantic units under different semantic fields, facilitating more comprehensive knowledge throughput towards the needs of knowledge reasoning. Besides, as the functional representations comprise explicitly defined relations among semantic units, *KasF* is highly flexible and sustainable, allowing users to edit the intrinsic structure explicitly in practical usage.

We first use the WikiQA dataset (Yang et al., 2015) to validate that the functional representations can capture more semantic patterns with an improved efficiency. Experiments show that *KasF* can efficiently encode knowledge with fewer parameters and better precision than traditional neural and SVD-based approaches. Next, we use the SQuAD2.0 benchmark (Rajpurkar et al., 2018) to validate the knowledge reasoning ability of *KasF* in the real-world case. We replace the last fully connected layers of several pre-trained language models fine-tuned on SQuAD2.0, *i.e.*, RoBERTa-base (Liu et al., 2019), ALBERTa-base (Lan et al., 2019), and DeBERTa-base (He et al., 2020), with our proposed *KasF* layers. Then we observe that the exact match (EM) accuracies of the *KasF*-based models outperform the original models by $0.95\% \sim 1.36\%$ with fewer parameters.

Furthermore, we choose CommonsenseQA (CSQA) Talmor et al. (2018) as a benchmark to validate *KasF*'s capability of complicated causal reasoning rather than just locating keywords in contexts. The CSQA's official team, evaluates our methods on a blind test set. The results show that the accuracy of *KasF* on a single model setting outperforms the previous best ensemble and single models by $0.4\%$ and $3.1\%$, respectively. Note that the prior best single model UnifiedQA (Raffel et al., 2020) is $47\times$ larger than ours. On the in-home developing set, the accuracy using a single fine-tuned ALBERT with *KasF* can be significantly increased from $78.7\%$ to $87.1\%$, outperforming models like GPT-3.5 (Bubeck et al., 2023) of $73.3\%$, and other fusion-based methods like ALBERT+KEAR of $81.2\%$ (Xu et al., 2021). Unlike models that use external task-specific resources, *e.g.*, large-scale QA training sets and a structured commonsense knowledge base like OMCS (Singh et al., 2002), *KasF* uses only the plain-text corpus and self-generative knowledge base during training and inference. Moreover, we show the advantage of *KasF* over main-stream Transformer encoders, demonstrate the feasibility of combining *KasF* with LLM decoder, and discuss the limitation regarding to implementation and application in the Appendix.

## 2 METHODOLOGY

### 2.1 FUNCTIONAL REPRESENTATION OF SEMANTIC UNITS

A semantic unit is analogous to a token/word and is strictly a conceptual element that implements reasoning tasks. A well-formed reasoning model needs as few semantic units as possible while preserving its reasoning capacity as much as possible. Giving a semantic unit with index $i \in \{1, ..., M\}$, Instead of treating it as a fixed-dimensional encoding, we treat it as a functional $\mathcal{F}_i$ that takes the functional and tensor-formed parameters, including the objectives $\mathbf{T}$, the external states $\mathbf{E}$, and the internal states $\mathbf{I}$ as the arguments, followed by returning a task-specific encoding $\mathbf{v}_i$ as the output:

$$\mathbf{v}_i = \mathcal{F}_i(\mathbf{T}, \mathbf{E}, \mathbf{I}) \tag{1}$$

The parameters in Eq. 1 have yet to be defined precisely. One can freely define the parameters towards the need of a given task, such as semantic compression, reading comprehension, and QA.

For better understanding, we show how to instantiate Eq. 1 to facilitate the task of semantic compression (Ceglarek, 2014; Cai et al., 2021). This task is to find the top-$K$ largest components of $z = \mathbf{V}y \in \mathbb{R}^{n \times 1}$ given a query vector $y \in \mathbb{R}^{D_v \times 1}$ and $n$ $D_v$-dimensional encodings $\mathbf{V} \in \mathbb{R}^{n \times D_v}$ under limited computational cost. In this case, the query $y$ is the external state $\mathbf{E}$ in Eq. 1. The internal states $\mathbf{I}$ contain two parts: the remaining computational budget and the original-dimensional encoding $\mathbf{V}_i \in \mathbb{R}^{D_v \times 1}$. The objective $\mathbf{T}$ minimizes the query loss between the computed $\hat{z}$ and the actual $z$, and the computational cost.

$$\mathbf{T}(D_v') = \sum_{k=1}^{K} \gamma^k \|y^\top \mathbf{V}_{r_k} - (\mathbf{P}[D_v']y)^\top (\mathbf{P}[D_v']\mathbf{V}_{r_k})\| + D_v'^{2} \tag{2}$$

where $D_v' \in [1, D_v - 1]$, $\gamma \in [0, 1]$, $\mathbf{P}[D_v'] \in \mathbb{R}^{D_v' \times D_v}$. The index $r_k$ refers to the candidate that corresponds to the $k$-largest components of $z$, i.e., $z_{r_k} = y^\top \mathbf{V}_{r_k}$ is the $k$-largest ones of $\{z_1, ..., z_n\}$. $\mathbf{P}[D_v']$ is trained by minimizing $\mathbf{T}(D_v')$ using a set of provided queries $y$. In this case, $\mathcal{F}_i$ presents a heuristic way to recursively compute $\mathbf{V}y$, reducing the dimensions step-by-step. During a $T$-step iterative process, $D_v^{(0)} = \frac{D_v}{10} < D_v^{(1)} < ... < D_v^{(T)}$, we compute the reduced $\mathbf{V}^{(t)}y^{(t)}$ and filter out the candidates, i.e., reduce the candidates' size from $R^{(t-1)} \in \mathbb{N}^+$ to $R^{(t)} \in [1, R^{(t-1)}]$, followed by further reducing the dimensions, computing and filtering out, etc., until the computational budget is consumed. The analytical form of $\mathcal{F}_i$ for this task can be defined as

$$\mathbf{v}_i^{(t)} = \mathcal{F}_i(\mathbf{T}, y, \mathbf{I}^{(t)}) = \mathbf{P}[D_v^{(t)}]\mathbf{v}_i^{(0)} = \mathbf{P}\Big[\eta\big(\Psi - \sum_{s=0}^{t-1} R^{(s)} D_v^{(s)^2}\big)^{\frac{1}{2}}\Big]\mathbf{v}_i^{(0)} \tag{3}$$

where $\mathbf{v}_i^{(0)} = \mathbf{V}_i$, $\eta \in (0, 1)$ is a predefined parameter that compromises the extra computational cost during pre-processing, and $\Psi \in \mathbb{N}$ is the initial computational budget that determines the expected computational complexity. The parameter $\mathbf{P}[D_v^{(t)}] \in \mathbb{R}^{D_v^{(t)} \times D_v}$ is pre-trained by back-propagating the objective $\mathbf{T}$ defined in Eq. 2. A candidate encoding $i$ with a higher $\hat{z}_i^{(t)}$ is more likely to be selected by the filtering mechanism, where details are omitted for simplicity. Note that this functional representation can perform specified tasks more efficiently, ensuring an optimal trade-off between the execution process's cost and efficiency. However, as illustrated above, this mechanism is too specific, it needs further generalized implementation.

### 2.2 GENERALIZING THE FUNCTIONAL REPRESENTATIONS OF KNOWLEDGE

The core idea involves the decomposition of a functional $\mathcal{F}_i$ into *local* and *global* functions. The local functions are responsible for processing the external inputs, referring to the external states $\mathbf{E}$ in Eq. 1 The global functions guide the local functions to achieve the ultimate outcomes. For analytical purposes, we assume that the reasoning model consists of $N$ neurons, each characterized by $d$-dimensional dynamic states as $q_x \in \mathbb{R}^d$ with $x \in \{1, ..., N\}$, following Pei & Wang (2023).

Subsequently, we will formalize the remaining arguments defined in Eq. 1. The objective argument $\mathbf{T}$ can be formalized as the task of nonlinear sequential mapping, as most reasoning tasks can be viewed as equivalent to such mappings. This task aims to map arbitrary sequential inputs, denoted

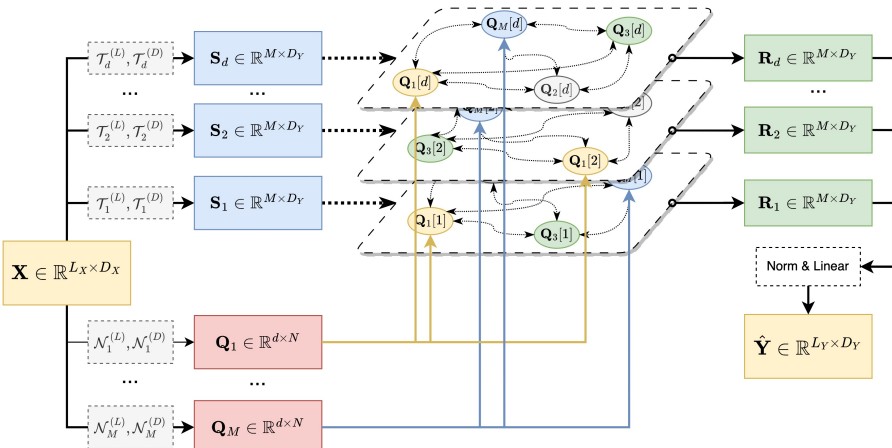

Figure 1: **A Sketch Pipeline of Functional Reasoning.** We associate the dynamic states corresponding to an identical semantic unit, *i.e.*, $\{\mathbf{Q}_i[1], ..., \mathbf{Q}_i[d]\}$, with arrows of the same color. Then, as labeled via dotted line, the signals $\{\mathbf{S}_1, ..., \mathbf{S}_d\}$ are processed with the relations amongst dynamic states, which act as an instruction to guide the signals $\mathbf{S}$, to generate the final outputs $\hat{\mathbf{Y}}$.

as $\mathbf{X} \in \mathbb{R}^{L_X \times D_X}$, where $L_X$ represents the length and $D_X$ represents the dimension, to arbitrary sequential outputs, denoted as $\mathbf{Y} \in \mathbb{R}^{L_Y \times D_Y}$.

Formalizing the internal states $\mathbf{I}$ requires insights from the field of neuroscience: the neuronal activities involve a superposition of $M$ distinct "frequencies" (Koudelková & Strmiska, 2018). Each frequency component corresponds to a semantic unit equipped with dynamic states represented as $\mathbf{Q}_i = [q_1, \dots, q_N] \in \mathbb{R}^{d \times N}$. We can employ global functions to decompose the external inputs $\mathbf{X}$ into $M$ sets of dynamic states. Then, we compute the relations among the decomposed dynamic states, referring to the internal states $\mathbf{I}$. These relations serve as instructions that guide the local functions in their treatment on the external inputs, resulting in the generation of the final outputs.

Next, we present how to construct the global and local functions analytically. We configure the semantic units by defining their dynamic states and signals from the inputs $\mathbf{X}$ via linear projections. As presented in Figure 1, we interpret $M$ semantic units as two pairs of $M$ global functions, $\{\mathcal{N}_i^{(L)}, \mathcal{N}_i^{(D)}, i \in [1, M]\}$, designed to deal with $\mathbf{X}$'s **L**ength and **D**imension, respectively. These global functions take the inputs $\mathbf{X}$ as the input argument and returns the functional representations of "knowledge" embedded in $\mathbf{X}$ as the dynamic states:

$$\mathbf{Q}_i = \mathcal{N}_i^{(L)} \mathbf{X} \mathcal{N}_i^{(D)} \in \mathbb{R}^{d \times N}; \; i \in \{1, ..., M\} \tag{4}$$

where $\mathcal{N}_i^{(L)} \in \mathbb{R}^{d \times L_X}$ and $\mathcal{N}_i^{(D)} \in \mathbb{R}^{D_X \times N}$ are trainable parameters corresponding to a specific functional $\mathcal{F}_i$. Likewise, we interpret the $d$-dimensional dynamic states as two pairs of $d$ local functions, *i.e.*, $\{\mathcal{T}_k^{(L)}, \mathcal{T}_k^{(D)}, k \in [1, d]\}$. These local functions also take $\mathbf{X}$ as the input argument, and returns $d$ distinct signals, *i.e.*, $\{\mathbf{S}_1, ..., \mathbf{S}_d\}$, for further processing. The signals $\mathbf{S}_k \in \mathbb{R}^{M \times D_Y}$ are computed via the local functions as follows:

$$\mathbf{S}_k = \mathcal{T}_k^{(L)} \mathbf{X} \mathcal{T}_k^{(D)} \in \mathbb{R}^{M \times D_Y}; \; k \in \{1, ..., d\} \tag{5}$$

where $\mathcal{T}^{(L)} \in \mathbb{R}^{M \times L_X}$ and $\mathcal{T}^{(D)} \in \mathbb{R}^{D_X \times D_Y}$ are trainable parameters shared by all the functionals $\mathcal{F} = \{\mathcal{F}_1, ..., \mathcal{F}_M\}$. The relations amongst the semantic units are obtained based on the piece-wise linearity principle, which indicates that an arbitrary nonlinear mapping can be approximated as the weighted sum of linear transformations. Thus, the $d$-dimensional nonlinear relation $\mu$ between two semantic units $i, j \in \{1, ..., M\}$ is interpreted as the weighted $p$-norm between dynamic states,

$$\mu_{ij}[k] = \sum_{h=1}^{H} \lambda_h \|\mathbf{Q}_i[k, h\delta : (h+1)\delta] - \mathbf{Q}_j[k, h\delta : (h+1)\delta]\|_p \in \mathbb{R}; \; \delta = \frac{d}{H} \tag{6}$$

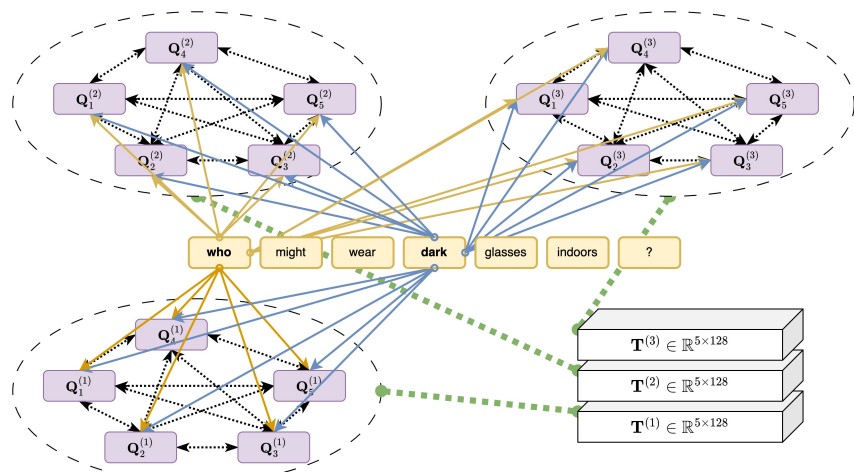

Figure 2: **Representation of a sentence using semantic fields.** The sentence contains 7 tokens interpreted as 128-dimensional embeddings, *i.e.*, $L_X = 7$ and $D_X = 128$, mapping to the hyperspaces via three distinct semantic fields, *i.e.*, $G = 3$. Each semantic field contains five semantic units, *i.e.*, $M = 5$. The generated encoding is a tensor of shape $\mathbb{R}^{G \times M \times D_X}$, as we suppose $D_X = D_Y$ here.

where $\lambda_h \in \mathbb{R}$ is a trainable parameter. Note that we use $p$-norm rather than other metric functions, such as cosine similarity, since $p$-norm results in a stronger representation capacity in empirical results. The resulting signals received by semantic units are obtained as

$$\mathbf{R}_k = \sigma[\sum_{m=1}^{M} \mu_{im}[k] \cdot \mathbf{S}_k[m,j]]_{i,j} = \sigma(\Phi_k \mathbf{S}_k) \in \mathbb{R}^{M \times D_Y} \tag{7}$$

where $\Phi_k \in \mathbb{R}^{M \times M}$ with $i \in \{1, ..., M\}$ and $j \in \{1, ..., D_Y\}$. Note that Eq. 7 can be largely accelerated via the designed matrix algorithms presented by Indyk & Silwal (2022) and Pei & Wang (2023). This faster implementation can compute Eq. 7 without explicitly knowing the matrix $\Phi_k$, reducing the computational complexity from $O(M^2 D_Y)$ to $O(pM D_Y)$, where $p$ refers to the $p$-norm in Eq. 6. The final output is then obtained as the weighted sum of resulting signals:

$$\hat{\mathbf{Y}} = \sum_{k=1}^{d} \mathbf{O}_k \mathbf{R}_k \in \mathbb{R}^{L_Y \times D_Y} \tag{8}$$

where $\mathbf{O}_k \in \mathbb{R}^{L_Y \times M}$ are trainable parameters shared by the functionals set $\mathcal{F}$. We can take either the MSE Loss or the Cross-Entropy Loss between $\hat{\mathbf{Y}}$ and $\mathbf{Y}$ as the objective function to train the trainable parameters mentioned in this section.

## 2.3 INTERPRETING KNOWLEDGE AS FUNCTIONAL VIA SEMANTIC FIELDS

Recall that a semantic field is a semantically structured group of the lexical set of words/tokens related to one another through their similar meanings or a more abstract semantic relation. Our proposed *KasF* interprets such an abstract semantic relation as a multi-dimensional nonlinear relation $\mu$ via Eq. 6. Suppose we have a sentence $S_X = [w_1, ..., w_{L_X}]$ with embeddings denoted as $\mathbf{X} = [\mathbf{v}_1, ..., \mathbf{v}_{L_X}]^\top \in \mathbb{R}^{L_X \times D_X}$. As shown in Fig. 2, *KasF* maps the embeddings into $d$-dimensional dynamic states of semantic units via Eq. 4. Note that a semantic unit is not strictly related to a single word but refers to a collection of words that constitute the semantics, so the semantic unit gains much stronger representation ability than the traditional distributed representation approach (Mikolov et al., 2013) and a massive body of its variants.

Besides, there should be multiple semantic fields, *i.e.*, $\mathbf{Q}_i^{(1)}, ..., \mathbf{Q}_i^{(G)}$, rather than just a single one from a linguistic view (Ataboyev & Turgunova, 2022). Each semantic field may refer to a particular linguistic pattern, *e.g.*, part-of-speech, dependency parsing, or even some unknown patterns. The

dynamical relations amongst the semantic units in a semantic field act as an instruction to guide the signals $\mathbf{S}^{(g)} = [\mathbf{S}_1^{(g)}, ..., \mathbf{S}_d^{(g)}]$, which are obtained from $\mathbf{X}$ via Eq. 5. Different from Eq. 8 that generates the final outputs $\mathbf{Y} \in \mathbb{R}^{L_Y \times D_Y}$, we obtain the temporal outputs by adding the resulting signals along with dimension, *i.e.*, $\mathbf{T}^{(g)} = \sum_{k=1}^{d} \mathbf{R}_k^{(g)} \in \mathbb{R}^{M \times D_Y}$.

Using the methods above, we can encode an arbitrary fixed-length sentence or query $S_X$ as a set of temporal signals denoted by $\mathbf{T}_X = \{\mathbf{T}^{(1)}, ...\mathbf{T}^{(G)}\}$, each refers to the dynamic states of a semantic field. For a reasoning task in the form of multiple choices and selecting the best choice from the candidates $\mathcal{C} = \mathcal{C}_1, ..., \mathcal{C}_C$, we need to get the temporal signals of all the well-formed sentences that contain the candidates. These well-formed base sentences are extracted from the training set and other relevant corpus. Then we use a trainable metric function $\mathcal{G}$ to compute the probability that a candidate $\mathcal{C}_x$ can answer the question $S_X$.

$$\mathcal{G}(\mathbf{T}_X, \mathbf{T}_{\mathcal{C}_y}) = \mathbf{W}^{(M)} \sigma \left( \sum_{g=1}^{G} \text{Concat}\left(\mathbf{T}_X^{(g)}, \mathbf{T}_{\mathcal{C}_y}^{(g)}\right) \mathbf{W}^{(D)} \right) \tag{9}$$

where $\mathbf{T}_{\mathcal{C}_y}$ refers to the temporal signals of a base sentence containing the candidate $\mathcal{C}_y$, $\mathbf{W}^{(M)} \in \mathbb{R}^{1 \times 2M}$, and $\mathbf{W}^{(D)} \in \mathbb{R}^{D_Y \times 1}$. We train the metric $\mathcal{G}$ by assigning it 1 for a correct candidate and $-1$ for an incorrect candidate.

## 3 EMPIRICAL RESULTS

### 3.1 DATASETS AND SETUPS

We use three public benchmarks, *i.e.*, WikiQA (Yang et al., 2015), SQuAD2.0 (Rajpurkar et al., 2018), and CommonsenseQA (Talmor et al., 2018), to validate our methods in knowledge reasoning tasks. We also use a synthetic dataset to compare *KasF* with Transformer (Vaswani et al., 2023).

The WikiQA dataset contains 3047 questions/queries and 29258 sentences/candidates, in which 1473 sentences were labeled as the answer to their related questions. Bing query logs were used as the query source to reflect the real-world case. Each query is linked to a Wikipedia article that contains the answer. We use the heuristic functional representation presented by Eq. 3 on this benchmark to show that the proposed functional representation contains more semantic patterns than other fixed-dimensional representations.

The SQuAD2.0 dataset is a challenging natural language understanding benchmark for reading comprehension. The dataset combines $100,000$ questions extracted from its former version with over $50,000$ new unanswerable questions written by crowdworkers. The training dataset contains 87k answerable and 43k unanswerable questions. In this task, we set the output's length the same as the input's length with a binary classifier that determines if the current token is the start or end token, *i.e.*, $L_Y = L_X$ and $D_Y = 2$. Besides, we set the numbers of semantic units and dynamic states $M = N = D_X$ as defaults. The default dynamic dimension $d$ is 30. We use some fine-tuned base models[3], *e.g.*, RoBERTa-base (Liu et al., 2019), ALBERTa-base (Lan et al., 2019), and DeBERTa-base (He et al., 2020), as the text encoders to obtain the encoded inputs feeding into the *KasF* module defined in Section 2.2, followed by fine-tuning them on the SQuAD2.0's training dataset.

The CommonsenseQA (CSQA) dataset is a challenging 5-way multiple choice QA dataset for commonsense question answering, containing $12,102$ questions created via ConceptNet (Speer et al., 2017). The official test set of CSQA is hidden, and the official team can only evaluate model predictions twice a month. Therefore, besides the official results in the leaderboard, we perform major experiments on the in-house data splits following Lin et al. (2019). We use RoBERTa-large (Liu et al., 2019) and ALBERTa-xxlarge-v2 (Lan et al., 2019) as the base models to build the pre-trained text encoders, respectively. Then, we feed the encoded inputs into a well-formed *KasF* module described in Section 2.3. The output mechanism follows the Eq. 9.

We also conduct experiments on a synthetic dataset based on the SQuAD2.0 dataset. The empirical results show that our *KasF* outperforms a typical transformer encoder in learning nonlinear sequential mapping. We refer the reader to the Appendix A.1 for more details.

---

[3]We select the best-performing models among the fully open-sourced and computational affordable models in the leaderboard. The models' structures are also relatively representative.

### 3.2 BASELINE METHODS FOR SEMANTIC COMPRESSION AND KNOWLEDGE REASONING

Table 1 lists the baseline methods for semantic compression. Incremental PCA, *i.e.*, **iPCA** (Evangelopoulos et al., 2012) reduces the query and candidate embeddings to lower dimensional compressed vectors. **Kernel PCA** (Mingbo et al., 2021) configures a non-linear mapping to transform the embeddings to higher-dimensional space, followed by standard PCA to project them back to lower-dimensional linearly separable space. Locally Linear Embedding, *i.e.*, **LLE**) aims to discover non-linear structures in the dataset and also preserve the distances within local neighborhoods (Roweis & Saul, 2000). Isometric Mapping, *i.e.*, **Isomap**, uses the spectral theory to preserve the geodesic distances in the lower dimensional space (Tenenbaum et al., 2000).

The baseline methods for knowledge reasoning are listed in Table 3, Table 4, and Table 5. **Fine-tuned LMs:** We add a linear layer with a softmax on the text encoder RoBERTa-large or ALBERTa-xxlarge-v2. The RoBERTa model follows the setting in Zhang et al. (2022). The ALBERTa model uses a learning rate of $1e^{-5}$, batch size 16, Adam optimizer, five epochs, and a linear learning rate scheduler. **GNN-based KGs:** The graph-based methods interpret commonsense rules as graph-based relational features, *e.g.*, KagNet (Lin et al., 2019), QA-GNN (Yasunaga et al., 2021), GreaseLM (Zhang et al., 2022) and HGN (Yan et al., 2020). These methods leverage the distributed representation of semantic components and several structure-based features. The topological relations amongst entities have a direct impact on the model predictions. **Fusing Mechanisms:** We also compare the methods that apply the fusing mechanism (*e.g.*, self-attention and external attention) to fuse contextual features into knowledge graphs, including DEKCOR (Xu et al., 2020), KEAR (Xu et al., 2021) and HeadHunter (Li et al., 2021). These methods combine the knowledge-centric features with the input textual encoding via novel attention-based mechanisms. The knowledge is expressed as triplets via a selection mechanism such as KCR (Xu et al., 2020), which uses the frequency of relation type to assign a weighted score to each triplet.

### 3.3 SEMANTIC COMPRESSION

Semantic search aims to improve search accuracy by capturing the semantic information of the content candidates. In Table 1, we choose three pre-trained models designed for semantic search, including *All-mpnet* (Song et al., 2020), *Qa-mpnet* (Song et al., 2020), and *Distil-roberta* (Sanh et al., 2019). to encode the queries and the candidate contents to generate contextual representations with the original dimension of 768 (Reimers & Gurevych, 2019). The experiments are conducted on four dimension settings, *i.e.*, $dim = \{10, 20, 30, 50\}$. The heuristic $KasF$ using Eq. 3 takes the original 768-dimensional embeddings as $\mathbf{v}^{(0)}$ defined in Eq. 3. We also list the actual time cost of our method for implementing semantic query to validate the fairness of comparison. *KasF* in all cases performs better than other methods regarding query accuracy. We observe that using the functional representation to encode the sentences, the contextual representations containing less than 10% of the original parameters perform competitively with the original 768-dim representations in semantic search. On text encoders of All-mpnet and Distil-roberta, our method with much fewer parameters and complexity performs even better than the original 768-dimensional representation obtained by a large-scale pretrained model, demonstrating the advantage of $KasF$ in encoding the relational information underneath the data.

### 3.4 READING COMPREHENSION

To implement reading comprehension via *KasF*, we need a text encoder to generate the inputs $X$. We use three distinct pre-trained language models as the text encoders in the experiments. Specifically, we replace their last output layers with our designed *KasF* module, then fine-tune the *KasF*-based models using the training corpus. As presented in Table 2, all the *KasF*-based language models outperform the original Linear layer with fewer parameters. The *KasF*-based models outperform those with a Transformer encoder as the output layer. These empirical results show *KasF* can capture more semantic patterns from the context. We further conduct experiments to compare a *KasF* module with a Transformer regarding the approximation capacity on sequences. We create a synthetic dataset using the sentences from SQuAD2.0 as the sequential inputs and randomly generated sequences as the labels. Then we observe that a *KasF* module can better fit the sequential mappings than a Transformer (Appendix A.1) with fewer parameters.

Table 1: **Performance comparison on WikiQA dataset.** We test our method on the task of semantic query on different dimension settings. Each dimension setting refers to a limited computational budget, *i.e.*, the time cost for implementing the complete query task should not exceed a specific value. We use the original embeddings generated by three distinct pre-trained text encoders. Then, we apply five methods to compress the generated embeddings or reduce the computational complexity to meet the limited computational budgets. The top 1 or top 3 accuracies are recorded.

| Compress dim | | dim=10 | dim=20 | dim=30 | dim=50 | | dim=768 | |
|---|---|---|---|---|---|---|---|---|
| Actual Time Cost for Query (seconds) | | 0.3-0.4 | 0.4-0.5 | 0.5-0.7 | 1.1-1.3 | | 12-13 | |
| Text Encoder | Method | Top1 | Top1 | Top1 | Top1 | Top3 | Top1 | Top3 |
| All-mpnet | iPCA | 0.0887 | 0.2759 | 0.3842 | 0.4926 | 0.7340 | | |
| | kPCA | 0.0690 | 0.2364 | 0.3941 | 0.5222 | 0.7438 | | |
| | LLE | 0.2167 | 0.2463 | 0.2562 | 0.2611 | 0.5025 | 0.5615 | 0.8374 |
| | Isomap | 0.2463 | 0.2562 | 0.2759 | 0.2906 | 0.5813 | | |
| | KasF (Ours) | **0.2611** | **0.3892** | **0.4384** | **0.5665** | **0.8079** | | |
| Qa-mpnet | iPCA | 0.0345 | 0.1330 | 0.2611 | 0.4433 | 0.6749 | | |
| | kPCA | 0.0246 | 0.1231 | 0.2512 | 0.4433 | 0.6601 | | |
| | LLE | 0.1674 | 0.1921 | 0.2019 | 0.1921 | 0.3744 | 0.6010 | 0.8276 |
| | Isomap | 0.1133 | 0.1133 | 0.1478 | 0.1724 | 0.3645 | | |
| | KasF (Ours) | **0.1872** | **0.3892** | **0.4828** | **0.5764** | **0.8177** | | |
| Distil-roberta | iPCA | 0.0493 | 0.2315 | 0.3399 | 0.3941 | 0.6749 | | |
| | kPCA | 0.0542 | 0.2213 | 0.3005 | 0.3892 | 0.6700 | | |
| | LLE | 0.1478 | 0.1823 | 0.1970 | 0.1773 | 0.3695 | 0.3990 | 0.7192 |
| | Isomap | 0.1773 | 0.2069 | 0.2118 | 0.2118 | 0.5074 | | |
| | KasF (Ours) | **0.2808** | **0.2709** | **0.3892** | **0.4089** | **0.6946** | | |

Table 2: **Performance comparison on SQuAD2.0 dataset**. We test our method for reading comprehension tasks by replacing the output layers of the selected fine-tuned pre-trained models. For each model, we use three configurations for *KasF*, denoting as the form of Gx-Dy, where $x$ refers to the number of semantic fields, and $y$ refers to the dimension of dynamic states. For further comparison, we use an additional Transformer encoder as the output layer for each model with 768-dimensional hidden size and 4 heads identically. The impromenvents validate the advantage of *KasF*.

| Base Model | Output Layer | No.Params | EM (%) | F1 (%) | Improvement (%) |
|---|---|---|---|---|---|
| ALBERTa-base | FC Linear | 0.59M | 75.82 | 78.91 | 0 |
| | Transformer | 2.98M | 76.25 | 79.46 | + 0.43 |
| | KasF (G4-D30) | 0.25M | 76.68 | 78.82 | + 0.86 |
| | KasF (G4-D50) | 0.41M | 77.18 | 80.31 | + 1.36 |
| RoBERTa-base | FC Linear | 0.59M | 79.68 | 82.24 | 0 |
| | Transformer | 2.98M | 80.13 | 82.76 | + 0.45 |
| | KasF (G4-D30) | 0.25M | 80.41 | 83.05 | + 0.73 |
| | KasF (G4-D50) | 0.41M | 80.63 | 83.22 | + 0.95 |
| DeBERTa-base | FC Linear | 0.59M | 81.20 | 84.25 | 0 |
| | Transformer | 2.98M | 81.82 | 84.93 | + 0.62 |
| | KasF (G4-D30) | 0.25M | 82.07 | 85.15 | + 0.87 |
| | KasF (G4-D50) | 0.41M | 82.44 | 85.54 | + 1.19 |

## 3.5 COMMONSENSE REASONING

Our results in Table 4 demonstrate substantial improvement over baseline models. For various text encoders, *KasF* consistently improves and significantly outperforms other approaches. We also submit our best single model to CSQA's team for evaluation on the official blind test set. Table 3 shows that *KasF* outperforms strong fine-tuned LM baselines (*e.g.*, RoBERTa-large and ALBERTa-xxlarge) and the best amongst all the single models. Specifically, UnifiedQA (Khashabi et al., 2020) has 11B parameters and is based on T5, which is impractical to fine-tune without strong GPU servers. ALBERT+HeadHunter uses Open Mind CommonSense corpus (Singh et al., 2002)

Table 3: **Test accuracy on CommonsenseQA's official leaderboard**. Note that models with * use ConceptNet. The CSQA's official team no longer accepts submission using ConceptNet. Our method outperforms all the prior ensemble and single models presented in leaderboard.

| Methods | Parameters | Test-Acc. (%) | | Use External Sources | |
|---|---|---|---|---|---|
| | | single | ensemble | QA datasets | KGs |
| BERT-large | $\sim 345M$ | 56.7 | - | - | - |
| KagNet* | $> 345M$ | - | 58.9 | ✓ | ✓ |
| RoBERTa-large | $\sim 354M$ | 72.1 | 72.5 | - | - |
| ALBERT-large | $\sim 223M$ | 73.5 | 76.5 | - | - |
| ALBERT+PathGenerator | $> 345M$ | 75.6 | 78.2 | - | ✓ |
| QA-GNN* | $\sim 360M$ | 76.1 | - | - | ✓ |
| ALBERT+HGN* | $\sim 355M$ | 77.3 | - | - | ✓ |
| T5 | $\geq 11B$ | 78.1 | - | - | - |
| ALBERT+HeadHunter | $\sim 283M$ | 78.4 | 78.3 | - | ✓ |
| UnifiedQA | $\sim 11B$ | 79.1 | - | ✓ | - |
| DeBERTa | $\sim 1.5B$ | - | 79.6 | ✓ | - |
| ALBERT+SFR | - | - | 81.8 | - | - |
| ALBERT+KasF (Ours) | $\sim 225M$ | **82.2** | - | - | - |

Table 4: **Performance comparison on CommonsenseQA in-house controlled experiments**. As the official test set is hidden, we report the accuracies of in-house dev (IHdev-Acc) and test (IHtest-Acc), following the data split of Lin et al. (2019). The DEKCOR* and KEAR* methods use the prohibited ConceptNet, whose empty triplets explicitly correspond to the human-generated distractor choices. Therefore, we randomly initiate the empty triplets to eliminate the shortcut hints.

| Methods | IHdev-Acc. (%) | IHtest-Acc. (%) |
|---|---|---|
| GPT-3.5-turbo | 73.3 | - |
| RoBERTa-large | $73.1 \pm 0.5$ | $68.7 \pm 0.6$ |
| +KagNet | $73.5 \pm 0.2$ | $69.0 \pm 0.8$ |
| +PathGenerator | - | $72.7 \pm 0.4$ |
| +QA-GNN | $76.5 \pm 0.2$ | $73.4 \pm 0.9$ |
| +HGN | - | $73.6 \pm 0.3$ |
| +KasF (Ours) | $\mathbf{79.5 \pm 0.3}$ | $\mathbf{75.4 \pm 0.4}$ |
| ALBERTa-large | $78.7 \pm 0.4$ | $75.1 \pm 0.4$ |
| +DEKCOR* | 80.3 | - |
| +KEAR* | 81.2 | - |
| +Headhunter | 83.3 | - |
| +KasF (Ours) | $\mathbf{87.1 \pm 0.3}$ | $\mathbf{83.8 \pm 0.4}$ |

as an additional knowledge base regarded as an extended commonsenseQA dataset. Our *KasF* that only uses self-generative resources still outperforms UnifiedQA and HeadHunter by 3.1% and 3.8%, respectively. The *KasF*'s superiority on complicated sentences is further validated in Appendix A.2.

## 4 CONCLUSION

We propose a Dynamics-inspired **K**nowledge **as F**unctionals mechanism for knowledge reasoning. We first evaluate *KasF* on the WikiQA dataset, demonstrating outstanding semantic compression and dimension reduction ability over other widely used paradigms. Then we evaluate *KasF* on the SQuAD2.0 dataset to validate its superiority in knowledge reasoning. Finally, we evaluate *KasF* on the official blind test set of CommonsenseQA, where *KasF* with single model setting achieves state-of-the-art in the leaderboard, outperforming the prior best ensemble and single models by 0.4% and 3.1%, respectively. Future works include systematic research on semantic fields and training a large-scale language model completely based on $KasF$ from scratch as both language encoder and decoder, and the framework that treats data and knowledge uniformly for constructing LLMs.

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

# A  APPENDIX

## A.1  COMPARISON WITH TRANSFORMERS

We constructed a synthetic dataset using sentences from the SQuAD2.0 dataset as inputs and generating 100-dimensional labels for each input token. We train our *KasF* module, as presented in Section 2.2, along with a standard Transformer encoder with 4 heads and 4 layers on this synthetic dataset. It is important to note that we intentionally adjusted the models to have fewer parameters, causing their representation capacities to be inadequate for fitting the training samples. This manipulation can measure the ultimate representation capacity of each model on a sequential dataset.

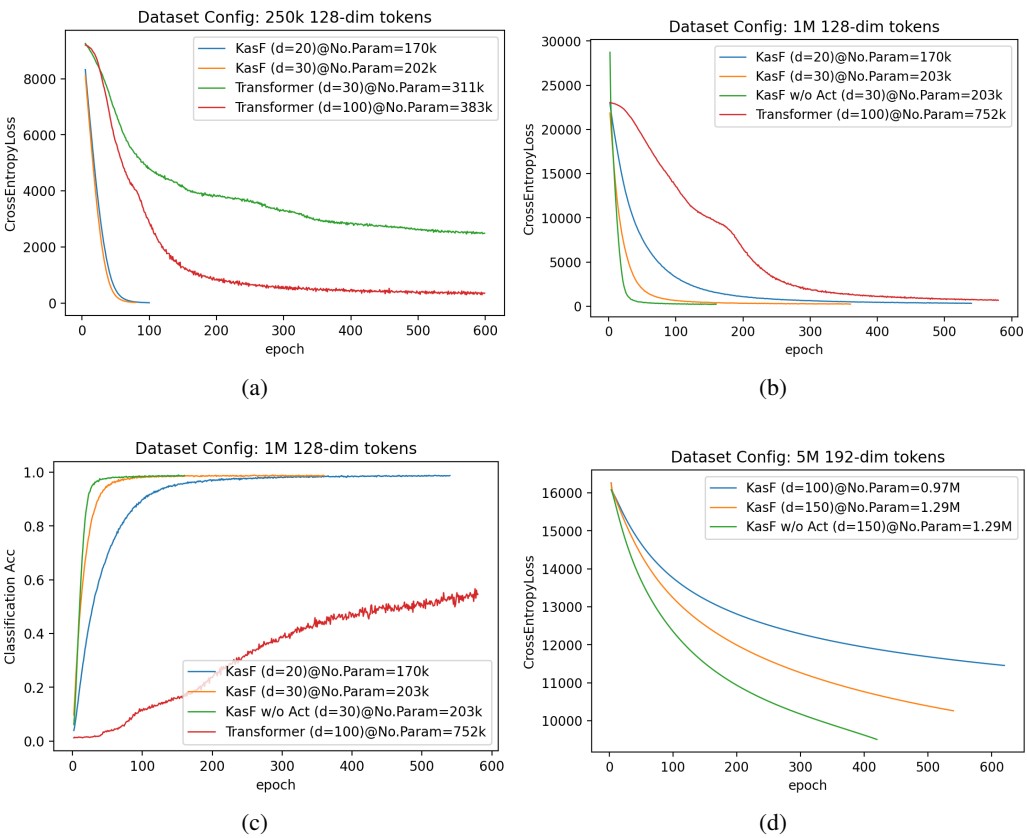

Figure 3: Comparison between *KasF* and Transformer Encoder.

As Figure 3 demonstrates, the *KasF* module performs better in fitting sequential samples with reduced parameter counts. Furthermore, as illustrated in Figure 3d, the *KasF* module with more parameters outperforms the others. It is worth emphasizing that *KasF* modules without activation functions (labeled as "w/o Act" in the figures) exhibit higher representation capacities. We postulate that this phenomenon occurs because the $p$-norm operations applied in Eq. 6 act as rectifiers, introducing nonlinearity to the model.

## A.2  EMPIRICAL RESULTS ON CSQA'S COMPLICATED SENTENCES

We investigate whether *KasF* makes consistent improvements in tasks requiring more complicated reasoning. We follow Zhang et al. (2022) to categorize the dev set into three proxies, *i.e.*, $a$) the number of prepositional phrases in the question stems, $b$) the existence of a negation term (*e.g.*, *no*, *not*), and $c$) the existence of a hedging term (*e.g.*, *possibly*, *probably*). We implement the data split via the spaCy toolkit (Vasiliev, 2020). Each token with a dependency relation *prep* or *neg* is labeled as a propositional or negation term, respectively. The results are presented in Table 5, where we see $KasF$ significantly outperforms other competitors in all the settings. The improvement can

be attributed to the trainable metric functions of *KasF* that capture the semantic relations amongst tokens. Therefore, $KasF$ succeeds in dealing with negation and hedge terms with more sensitive meanings than the others.

Table 5: **Performance comparison on CommonsenseQA IHdev set to validate *KasF*'s improvements on complex questions.** The experimental setting follows Zhang et al. (2022). Except for the base models, the accuracy improvements are listed on the right side of the accuracy.

| Methods | No. Prepositional Phrases | | | | | | | | | | Negation Term | | Hedge Term | |
|---|---|---|---|---|---|---|---|---|---|---|---|---|---|---|
| | 0 | | 1 | | 2 | | 3 | | 4~7 | | | | | |
| RoBERTa | 66.7 | | 72.3 | | 76.3 | | 74.3 | | 69.5 | | 63.8 | | 70.7 | |
| +QA-GNN | 76.7 | +10.0 | 76.2 | +3.9 | 79.1 | +2.8 | 74.9 | +0.6 | 81.4 | +10.9 | 66.2 | +2.4 | 76.0 | +5.3 |
| +GreaseLM | 75.7 | +9.0 | 79.3 | +7.0 | 80.4 | +4.1 | 77.2 | +2.9 | 84.7 | +15.2 | 69.9 | +6.1 | 78.4 | +7.7 |
| +KasF (Ours) | **77.9** | **+11.2** | **79.8** | **+7.5** | **81.5** | **+5.2** | **80.6** | **+6.6** | **85.0** | **+15.5** | **81.5** | **+7.5** | **78.9** | **+8.2** |
| ALBERTa | 73.9 | | 77.8 | | 78.7 | | 75.0 | | 78.9 | | 81.4 | | 70.8 | |
| +KEAR | 80.6 | +6.7 | 82.8 | +5.0 | 80.1 | +1.4 | 80.6 | +5.6 | 84.2 | +5.3 | 86.6 | +5.2 | 73.9 | +3.1 |
| +KasF (Ours) | **83.8** | **+9.9** | **83.6** | **+5.8** | **84.8** | **+6.1** | **86.1** | **+11.1** | **94.7** | **+15.8** | **88.7** | **+7.3** | **79.3** | **+8.5** |

## A.3 MATHEMATICAL VALIDATION

Theoretically, the metric function $\mathcal{G}$ defined in Eq. 9 is dense in $C(I_d)$ that denotes the space of continuous functions on a finite $d$-dimensional cube (Theorem A.1). Thus, provided with a specific set of semantic fields $\mathcal{N}$, a well-formed prompt $\mathbf{S}$, and a discriminatory metric function $\mu$, our *KasF* can approximate arbitrary function amongst semantic units. This theorem guarantees that *KasF* can facilitate the usage of knowledge as nonlinear and dynamical functions. It also validates that the nonlinear dynamic functions of *KasF* can be of arbitrarily large model capacity with an arbitrary size of trainable parameters, supporting the full utilization of arbitrary-scale knowledge bases. Moreover, we can replace a query token with a proper candidate that provides sufficient information from arbitrarily large external knowledge sources.

**Theorem A.1.** *Let $\mathcal{V}$ be a $d$-dimensional vector space, $\Psi : \mathbb{R}^d \times \mathbb{R}^d \to \mathbb{R}^d$ constructed using a functional $\mathcal{F}$ and functions $\mathcal{N}$ defined above, be a continuous and discriminatory function on $I_d$, and $f_i : \mathbb{R}^d \to \mathbb{R}^{N \times d}$ be a vector field for $u_i \in \mathcal{V}$. Then there exists a signed regular Borel measure $\mu$ on $I_d$ such that $f_i(\theta) + f_j(-\theta) = \mu(u_i - u_j)$ for arbitrary $\theta \in \mathbb{R}^d$ and $S$ is dense in $C(I_d)$.*

*Proof.* (sketch) We can easily construct a well-formed measure $\mu$. We can prove in an apagogical manner that $\Psi$ is dense in $C(I_d)$. Suppose that $\Psi$ is not dense in $C(I_d)$, then the closure of $\Psi$ is a subset of $C(I_d)$: $\overline{\Psi} \subset C(I_d)$. By Hahn-Banach Theorem, there exists a non-zero bounded linear functional $\mathscr{L}$ on $C(I_d)$ such that $\mathscr{L}(\Psi) = \mathscr{L}(\overline{\Psi}) = 0$. By Reisz Representation Theorem, we have

$$\mathscr{L}(\Psi) = \int_\theta \Psi\Big(f_i(\theta), f_j(\theta)\Big) d\mu(\theta) = 0 \tag{10}$$

for every pair of $u_i$ and $u_j$. Since $\Psi$ is discriminatory, we must have $\mu = 0$, which contradicts the assertions that $\mathscr{L} \neq 0$ and $\mu$ are regular. $\square$

## A.4 A PRELIMINIARY EXPERIMENT ON COMBINING *KasF* WITH LLM

| | CommonsenseQA | PIQA | Social iQA |
|---|---|---|---|
| LLaMA2-7B | 57.8 | 78.8 | 48.3 |
| LLaMA2-7B+KasF | 59.2 | 80.0 | 49.4 |
| LLaMA2-13B | 67.3 | 80.5 | 50.3 |
| LLaMA2-13B+KasF | 69.3 | 81.1 | 51.5 |

Table 6: The results of LLMs with our *KasF* module

Our *KasF* can be seen as a new paradigm to model the relation among semantic units. Since it can be used to replace the Transformer encoder as shown in Appendix A.1, it is also feasible to be injected into the Transformer decoder of LLMs. To further validate the potential of our method to combine with LLMs, we have expanded our experimentation by utilizing the pre-trained LLaMA2-7B and LLaMA2-13B models on three distinct reasoning benchmarks: CommonsenseQA, SocialiQA and PiQA, see Table 6. Due to limited computational resources, we opted for a pragmatic approach, replacing the output linear layer (e.g., 4096×32000) with our proposed *KasF* structure. Subsequently, we fine-tuned the added KasF on each benchmark's training set while keeping the LLaMA's parameters fixed. Our results demonstrate consistent improvements in accuracy on the validation sets.

It is worth noting that the reasoning ability can potentially be further enhanced, provided richer computational resources and access to external knowledge bases, without imposing constraints on fixing the LLaMA's parameters. The results show that our *KasF* can also work with LLMs in decoder style, and enhance the accuracy accordingly. However, from the experiments, especially the results on CommonsenseQA validation dataset, we can observe that, the GPT-based LLMs, even with *KasF*, do not perform quite well on the commonsense reasoning task in multi-choice form, compared to the results obtained by using ALBERT in the main paper. However, it should be noticed that improving the GPT-based LLM's performance is beyond the scope of this paper.

## A.5 DISCUSSION ON THE POTENTIAL OF EMPLOYING EXTERNAL KNOWLEDGE BASES

Based on the definition of *KasF*, we have validated its ability in modeling the semantic unit relations in sentences. Another advantage of *KasF* is that it is naturally compatible on the task of encoding external knowledge bases. To elaborate, one can establish a semantic field in the form of a knowledge graph by substituting the dynamic state $Q$ of semantic units with embeddings of entities from a knowledge graph. Following this, we change the metric between $Q$s as the vector of relations between entities, creating an independent semantic field detached from the input $X$. In our preliminary experiments conducted on CsQA, we noted that adopting the mentioned approach to convey knowledge related to the query using corresponding triplets in ConceptNet led to an improvement of approximately 3.5% in model accuracy. Nevertheless, due to the evaluation protocol of CsQA that ConceptNet is not allowed to use, we have not included this part of results in the main part.

Furthermore, our findings indicate that by ensuring precision and accuracy in the independent semantic field through manual annotation, the model accuracy could be enhanced to over 94%, showing the significant potential for an enhanced $KasF$ through integration with well-established external knowledge bases.

## A.6 DISCUSSION ON THE LIMITATION OF *KasF*

We would like to discuss briefly on the limitation of our *KasF*, particularly regarding to the GPU implementation and hyper-parameter tuning issues:

- CUDA Implementation. One challenge we face is the limited availability of well-established CUDA packages for efficiently computing metric-based operations that are intensively required by our method. Integrating our approach into the Large Language Model (LLM) community demands additional efforts, particularly in terms of CUDA implementation and acceleration.

- Hyper-parameter Tuning. The introduction of novel concepts in our method, such as neuronal groups, represents an area that is not extensively explored in the existing literature. Consequently, further investigations are needed to empirically establish the relationships between these concepts and their theoretical implications. We acknowledge the importance of refining the content regarding to these novel elements and will address this concern in the revision by providing additional insights and analyses.

