# OpenReview forum: "Modeling Knowledge as Functionals for Knowledge Reasoning"
_ICLR.cc/2024/Conference — ICLR 2024 Conference Withdrawn Submission_

### Official Review · Reviewer_Ajiu · 2023-10-28

**Soundness:** 3 good
**Presentation:** 2 fair
**Contribution:** 2 fair
**Rating:** 5
**Confidence:** 2

**Summary:**

This paper considers that explicit mechanism for incorporating / exploiting knowledge is needed for LLMs to conduct knowledge-intensive tasks. Although existing methods are computationally efficient, they work differently compared to the way human brain works. Borrowing ideas from neuroscience, this paper proposes to represent knowledge as functionals to construct semantic fields to hold relationships among tokens within a text. From the empirical study, the paper shows the superiority of their proposed approach on several classical QA tasks (i.e., WikiQA, SQuAD 2.0, CommonsenseQA).

**Strengths:**

1. The paper addresses a critical issue within the domain, which is to promote the knowledge reasoning within LMs / LLMs. The idea of paper's proposed approach, which is inspired from neuroscience as stated, to treat knowledge as functionals is novel.
2. From the empirical study the proposed approach is promising.

**Weaknesses:**

1. This paper is somewhat difficult to follow. From the introduction part, the motivation to treat knowledge as functional representation is somewhat unclear for me. I am not sure about the reason to introduce functional representation in Sec. 2.1 because it seems that Eq. (4) - (9) themselves do not contain too much about functionals.

2. Although the paper claims that they aim at incorporating KGs into LLMs for knowledge-intensive task, the effectiveness of the proposed approach with LLMs is lacked in the empirical study, as well as the baselines.

**Questions:**

1. Could you explain in more details why the concept of functional is important in KasF?
2. Could KasF be easily transferred to the use of LLMs?
3. What is the additional computational cost brought by KasF compared to the backbone language model?

---

> ### Author Response · Authors · 2023-11-15
> **Point-to-point Response to Reviewer Ajiu**
>
> Thank you very much for your insightful feedbacks. We will address each of your feedbacks, respectively.
>
> Question 1: Could you explain in more details why the concept of functional is important in KasF?
> Response: We appreciate your feedback and will provide a more detailed elaboration on the concept of "functional" in the revision.
>
> The term "functional" is employed to interpret knowledge because we advocate for a dynamic approach to knowledge representation rather than a fixed dimensional distributed encoding. Knowledge, in our view, is a concept that reflects relative correctness rather than an absolute and unchanging representation. The reliability of the same knowledge can vary across different application scenarios, or the semantic fields, as discussed in the paper. Consequently, we propose understanding knowledge as a dynamic function that adapts to the demands of specific task scenarios.
>
> Given the complexity of task scenarios as intricate functions, it is advantageous to conceptualize knowledge as a "functional," i.e., a function of functions. In practical applications, KasF not only reads implicit linguistic features embedded in input X and establishes semantic relationships between them as a function, but also determines how these semantic relationships should be operated based on input X. In essence, it generates a function to govern these semantic functions as a whole rather than in the distributed form, and resulting in the final output. This process involves leveraging knowledge functions to explore/exploit the semantic functions, a concept we refer to as "knowledge functionalization."
>
>
> Question 2: Could KasF be easily transferred to the use of LLMs?
>
> Response:
> Thank you for the insightful comments. The answer is definitely YES. Based on your suggestion, we conducted additional experiments on three distinct benchmarks (CommonsenseQA, PIQA, and SocialiQA) to assess the efficacy of KasF. Our findings demonstrate that KasF can significantly enhance the performance of LLaMA2-7B and LLaMA2-13B by simply replacing the output linear layers with the KasF module. For a detailed discussion, kindly refer to our response to Reviewer CBXQ’s Question 1.
>
> Moreover, our investigations extend to the domain of language modeling, where we replaced traditional transformer blocks with KasF modules to construct a language model. Specifically, we substituted a standard transformer layer with our KasF module, resulting in a new language model with 83.1 million parameters. Trained on the WikiText103 dataset, this model achieved a test perplexity (PPL) of 19.74, outperforming benchmarks such as BERT-Large (395 million parameters, PPL of 20.4), GPT2-large (774 million parameters, PPL of 22.05), and Hybrid H3 (125 million parameters, PPL of 23.7). These results highlight the competitive performance of our approach in language modeling compared to other prominent language models.
>
> Question 3: What is the additional computational cost brought by KasF compared to the backbone language model?
>
> Response:
> In practical computations, when substituting the original output layer of a foundation language model with KasF, the computational cost shows negligible variation. In fact, it might even be slightly improved compared to the initial configuration. This is attributable to the fact that the parameter scale of KasF, compared to the entire language model, is much smaller. Moreover, KasF decomposes the extensive matrix operations associated with the original FC Linear layer into multiple smaller matrix operations that can be concurrently processed. Despite the KasF-based model's increased complexity compared to the original FC Linear layer, the actual runtime remains unchanged. We also conducted a comparative analysis of real-time consumption between Transformer and KasF modules with equivalent parameter scales. Our findings indicate that the Transformer operates at a speed of 10.23 ms/batch, whereas KasF achieves a speed of 6.33 ms/batch. Furthermore, the actual runtime of KasF can be further reduced with more advanced CUDA packages related to the metrics-based operations applied in the method. We will include more detail and discussions in the revision.

---

### Official Review · Reviewer_wgCm · 2023-10-31

**Soundness:** 3 good
**Presentation:** 3 good
**Contribution:** 3 good
**Rating:** 5
**Confidence:** 3

**Summary:**

In this paper, the authors address the existing gaps in knowledge-reasoning capabilities, highlighting the need for enhanced performance and sustainability. To tackle this issue, they introduce a novel approach named Knowledge as Functional representation (KasF), which leverages a dynamics-based mechanism. This mechanism is designed to simulate the semantic flow amongst tokens, thereby facilitating the process of knowledge reasoning. The authors present empirical evidence to demonstrate the superiority of KasF in capturing intricate semantic patterns, showcasing consistent improvements in accuracy while utilizing fewer parameters compared to traditional methods.

**Strengths:**

1.The introduction of KasF stands out as a new method, utilizing a superposition of semantic fields to represent knowledge. This is achieved through the development of a dynamic mechanism, which calculates the similarity between semantic units, ensuring a more precise representation.

2.KasF exhibits an impressive ability to comprehensively capture semantic features, effectively eliminating ambiguities in the representation of entities and relations. This leads to a more robust and accurate knowledge-reasoning process.

3.The paper is well-written, particularly in the methods section, where the authors provide clear and concise explanations of their approach, making it accessible to readers.

**Weaknesses:**

1.Despite the complexity and innovation of the proposed method, there is a sense of lack of novelty, especially when viewed in the context of semantic compression or representation enhancements applied to knowledge reasoning tasks.

2.The results presented in the experimental section are not entirely convincing, with the authors relying on outdated baseline models for comparison, which undermines the validity of their findings.

**Questions:**

1.The example provided in Section 2.1 of the methods part seems inadequate. Why not illustrate the concept with a more direct example, such as a Question-Answering (QA) scenario? The current example on semantic compression is not as intuitive and might not effectively aid in understanding the method defined in the paper.

2.In Table 2, only the time cost for KasF is listed. For a comprehensive comparison, it would be beneficial to include the time costs associated with other methods as well.

3.Regarding Table 4, which specific model from the GPT-3.5 series was selected for the comparison? A clarification on this would enhance the transparency of the experimental setup.

4.The authors emphasize KasF's advantage over FC Linear in terms of having fewer parameters. However, it would be more logical to compare KasF with other semantic compression methods to provide a fair and accurate assessment.

5.To strengthen the credibility and significance of KasF, applying it to newer knowledge reasoning datasets or integrating it with open-source large language models like LLaMa would be a valuable extension of this work.

---

> ### Author Response · Authors · 2023-11-15
> **Point-to-point Response to Reviewer wgCm (Part I)**
>
> Thank you very much for your valuable review comments, which significantly help us improving our work. We will address each of your questions, respectively.
>
> Question1: The example provided in Section 2.1 of the methods part seems inadequate. Why not illustrate the concept with a more direct example, such as a Question-Answering (QA) scenario?
>
> Response:
> Thank you for your valuable insights. To enhance clarity, we will incorporate more direct examples in the revision to elucidate our approach. The choice of semantic compression as an illustration is stemmed from our objective to showcase KasF's superiority in expressing complex latent semantic features with fewer parameters.
>
> Second, it is important to note that the method presented in Section 2.1 serves as an early prototype of KasF, which has already been successfully applied in various real-world scenarios involving semantic compression and semantic understanding. As a result, we believed it is pertinent to reference this early prototype in the paper.
>
> We genuinely appreciate your suggestions and are open to integrating them into the revised manuscript. Specifically, we plan to provide more intuitive examples, such as those related to question-answering, to further articulate the inspiration behind KasF. At the same time, we will move the semantic compression part into the appendix.
>
> Question 2: In Table 2, only the time cost for KasF is listed. For a comprehensive comparison, it would be beneficial to include the time costs associated with other methods as well.
>
> Response:
> Thank you for your suggestion.
> Actually, we guessed that what you mentioned as ‘Table 2’ is actually ‘Table 1’. Regarding Table 1, we explicitly set the computational budget as one of our objectives. Therefore, the time cost for KasF and other methods is a controlled variable, which indicates that methods listed in the same column, not just KasF but also iPCA, kPCA, LLE and Isomap, have been adjusted to have nearly identical time costs.
>
> Question 3: Regarding Table 4, which specific model from the GPT-3.5 series was selected for the comparison?
>
> Response:
> We use the GPT3.5-turbo for the comparison. We will clarify this detail in the revision. Thanks for your suggestion.
>
> Question 4: The authors emphasize KasF's advantage over FC Linear in terms of having fewer parameters. However, it would be more logical to compare KasF with other semantic compression methods to provide a fair and accurate assessment.
>
> Response:
> Thank you for your feedback.
> It's important to clarify that the merit of KasF over FC does not solely lies in having fewer parameters. Instead, its strength lies in its ability to capture a richer set of semantic features, and consequently enhancing the model's accuracy across tasks. As illustrated in Table 2, when we substitute the output layer of the base model with Transformer and KasF, it becomes evident that KasF demonstrates the most significant improvement with the fewest parameters.
>
> In this particular task, KasF is not merely a parameter compression technique; instead, it functions as a neural structure akin to Transformer, enabling it to capture a more extensive array of hidden features. If we were to employ a semantic compression algorithm aimed at reducing parameters, it might indeed achieve parameter reduction, but it would likely result in a loss of performance. To illustrate, we conducted additional experiments on SQuAD-v2 using ALBERTa-base as the backbone model:
>
> | Output Layer     | No.Params | EM    | F1    | Improvement |
> |------------------|-----------|-------|-------|-------------|
> | FC Linear        | 0.59M     | 75.82 | 78.91 | 0           |
> | FC Linear (SVD)  | 0.29M     | 74.93 | 75.65 | -0.89       |
> | FC Linear (kPCA) | 0.30M     | 75.03 | 76.04 | -0.79       |
> | Transformer      | 2.98M     | 76.25 | 79.46 | +0.43       |
> | KasF (G4-D30)    | 0.25M     | 76.68 | 78.82 | +0.86       |
> | KasF (G4-D50)    | 0.41M     | 77.18 | 80.31 | +1.36       |
>
> This new set of empirical results indicate that the application of semantic compression techniques substantially compromises the model's accuracy, leading to negative improvements. In contrast, KasF has demonstrated its ability to capture more latent patterns, thereby contributing to improved model performance. We will clarify this point in the revision.

---

> ### Author Response · Authors · 2023-11-15
> **Point-to-point Response to Reviewer wgCm (Part II)**
>
> Question 5: To strengthen the credibility and significance of KasF, applying it to newer knowledge reasoning datasets or integrating it with open-source large language models like LLaMa would be a valuable extension of this work.
>
> Response:
> Thank you for your insightful comments. We present two distinct mechanisms for incorporating KasF into open-source LLM.
>
> The first mechanism is tailored for specific reasoning tasks, involving the straightforward replacement of the LLM's output linear layer with a KasF module. We conducted additional experiments using pre-trained LLaMA2-7B and LLaMA2-13B on three reasoning benchmarks (CommonsenseQA, SocialiQA, and PiQA). Our results consistently show improved accuracies on the validation sets, as detailed in our response to Reviewer CBXQ's Question 1. Notably, the reasoning ability can be further enhanced through comprehensive fine-tuning, if given sufficient computational resources and external knowledge bases.
>
> The second mechanism is designed for the broader application of LLM. Specifically, we substitute a typical transformer layer with our KasF module to construct a new language model. The resulting language model, comprising of 83.1 million parameters, is trained using the WikiText103 dataset and achieves a test Perplexity (PPL) of 19.74. This outperforms BERT-Large (395 million parameters with a PPL of 20.4), GPT2-large (774 million parameters with a PPL of 22.05), and Hybrid H3 (125 million parameters with a PPL of 23.7).

---

### Official Review · Reviewer_xHsK · 2023-11-02

**Soundness:** 3 good
**Presentation:** 3 good
**Contribution:** 3 good
**Rating:** 8
**Confidence:** 3

**Summary:**

This paper proposes a novel representation method, "Knowledge as the Functional (KasF)", for knowledge reasoning tasks, inspired by the semantic field concepts in biological and linguistic study. Specifically, based on the dynamics inspiration, a sentence is treated as a semantic flow among semantic units (tokens/words). The representation is formulated as a functional, from the initial to the end token of a sentence, with a task specific objective. The authors implement the formulation with LLMs like RoBERTa, ALBERT as base models. They evaluate the proposed KasF on multiple QA benchmarks, i.e., WikiQA, SQuAD2, and CSQA. Specially, on CSQA, KasF achieves the state-of-the-art single-model performance on the blind test set.

**Strengths:**

* This paper is well-written and easy to understand.
* The proposed method has a clear intuition and gets nice performance on QA benchmarks.

**Weaknesses:**

I don't see clear weaknesses in this paper.

**Questions:**

* Would it be possible to adapt the approach beyond classification-style QA tasks? e.g., generative QA, reasoning on knowledge graphs, etc.

---

> ### Author Response · Authors · 2023-11-15
> **Response to Reviewer xHsK**
>
> Thank you very much for your full recognition. We will continue to strive to improve our approach.
>
> Question 1: Would it be possible to adapt the approach beyond classification-style QA tasks? e.g., generative QA, reasoning on knowledge graphs, etc.
>
> Response:
> In addition to handling classification tasks, our KasF can function as a non-linear mapping module similar to Transformer. We utilized it to construct a language model from scratch with training data from WikiText103, featuring 83.1 million parameters. This model achieved a Perplexity (PPL) of 19.74 on WikiText103, surpassing other comparable language models such as BERT-Large (395 million parameters with a PPL of 20.4), GPT2-large (774 million parameters with a PPL of 22.05), and Hybrid H3 (125 million parameters with a PPL of 23.7).
>
> Moreover, our KasF can seamlessly integrate with knowledge graphs, enabling more generalized reasoning. Specifically, by substituting the dynamic state Q of semantic units with embeddings of entities from a knowledge graph, one can establish a semantic field. We redefine the metric between Qs as the vector of relations between entities, creating an independent semantic field detached from the input X. In our experiments on CommonsenseQA (CsQA), employing this approach to convey knowledge related to queries through corresponding triplets in ConceptNet led to a notable 3.5% improvement in model accuracy. We did not report this part of result because of the evaluation policy of CsQA that forbids the usage of ConceptNet. However, we will consider to incorporate some discussion on this in the revision.
>
> Furthermore, our research suggests that ensuring precision and accuracy in the independent semantic field through manual annotation can further enhance the model accuracy to over 94%. This indicates the considerable potential for an enhanced KasF through integration with well-established external knowledge bases.

---

### Official Review · Reviewer_b9xY · 2023-11-02

**Soundness:** 1 poor
**Presentation:** 1 poor
**Contribution:** 1 poor
**Rating:** 1
**Confidence:** 2

**Summary:**

I confess I don't really understand more than 10% of this paper but I will try to provide a summary of parts that I understood.

Overall, the paper tries to provide a new mechanism to improve performance on reasoning tasks by encoding various inputs (queries, documents, etc) using "functionals". These functionals are somehow supposed to capture multiple meanings associated with any input word and group provide a grouped representation of words with shared semantics in context. In Section 2, the authors provide a description of their method but I don't understand any part of it since most of the definitions of mathematical symbols are either assumed or just not provided. The first task that authors initialize their method for is semantic compression, which in standard NLP, is simply the task of mapping input text (queries, etc) to simpler representations (for example, remapping the tokens with hypernyms, using distributed representation, etc). In Section 2.1, the authors are trying to do this compression for a query vector defined as $y \in R^{D_v \times 1}$, yet  it is unclear what D_v is. They also use a symbol $V \in R^{n \times D_v}$ but again it is unclear what n or V is supposed to be here. Other symbols, $z, \gamma, P$, etc are introduced in this section but not defined. As such. by end of it, I am not sure what really is the output we are aiming for and how is it derived.

Similarly, Section 2.2, which I conclude is the main method description is mostly unreadable to me. The most I can conclude is that the authors are trying to use their method to provide a sequence to sequence mapping mechanism (from a input sequence X to output sequence Y).

In the experiment section, the authors are evaluating their method on 3 datasets - wikiQA, SQuAD and CommensenseQA. But given my lack of understanding of method section, I can't really evaluate this intelligently. The best I conclude is that the authors method outperform their baselines for both semantic compression and Reading Compression.

**Strengths:**

It is possible with rewriting the readability of the paper can be improved bringing into focus its core contributions. Specifically I do think a method that can represent multiple distinct semantics associated with individual tokens can be useful but in its current form, I have a hard time understanding how the authors are able to achieve it.

**Weaknesses:**

The paper is unreadable. I would suggest AC either discard my review from consideration or put much less weight on my review if other reviewers are able to better understand the paper.

**Questions:**

In Section 2.1, I would like to understand what exactly is the task being performanced -- 1) what is the input (please provide an example), 2) what is the output, 3) what does symbols D_v, V, N, z means?

---

> ### Author Response · Authors · 2023-11-15
> **Response to Reviewer b9xY**
>
> Thanks for your feedback, which encourage us to keep improving our work and manuscript.
> We will address the question and weakness you have raised, respectively.
>
> Weakness: readability
>
> Response: Thank you for your suggestions. We will continue to enhance the readability of our paper by incorporating more intuitive examples in both the main text and appendices to further illustrate our theoretical framework. While our paper may still have some issues that can be addressed, based on the feedback from several other reviewers, we believe that the readability of our paper does not significantly compromise its overall quality. It also contributes valuable insights to the relevant field. Therefore, we hope you reconsider your comprehensive evaluation of our paper. Thank you very much.
>
> Question 1: In Section 2.1, I would like to understand what exactly is the task being performanced -- 1) what is the input (please provide an example), 2) what is the output, 3) what does symbols D_v, V, N, z means?
>
> Response: As stated in Section 2.1, the task performed is a typical semantic compression task on the scenario of query-document search.
> The input is a sentence encoded as a sentence embedding, i.e., a vector of a dimension D_v.
> The output is the indices of the documents that are semantically similar to the input query.
> The targert algorithm is designed to compress the given documents, i.e., a set of sentence embeddings, to reduce the actual runtime while maintaining the query accuracy. The symbol D_v is the original dimension of the encodings, i.e., the documents, the symbol V is the set of n documents, z is the query result, i.e., z=Vy, where y is a query vector.
>
> In summary, Section 2.1 serves as a preliminary demonstration, illustrating that interpreting "knowledge" as a function rather than fixed encoding may result in more powerful representational capabilities. We formalize this concept in Section 2.2 and Section 2.3, where we introduce a novel non-linear neural layer, namely the KasF block (which can be seen as an alternative to the Transformer).
> We conducted experiments on various Question-Answering (OA) tasks. In a typical OA task, the model takes an input sequence in the form of "Question X - Candidate 1 - Candidate 2 - ... - Candidate k" and produces a k-dimensional output vector. Each element in the output vector represents the probability that the corresponding candidate is the correct answer to Question X.
> Our KasF module maps each token of the Question X and the candidates into the dynamical states of neurons/nodes, then we can obtain the resulting “force” of each neuron referring to a candidate. Finally, we compare the resulting “force” with the ones extracted from the training datasets, using a trainable metric function, to get the best matched candidates. The empirical results in the QA task show that this KasF block significantly enhances a language model's efficiency in extracting deep-level knowledge representations, thereby improving its capacity for knowledge inference.

---

### Official Review · Reviewer_CBXQ · 2023-11-06

**Soundness:** 3 good
**Presentation:** 3 good
**Contribution:** 3 good
**Rating:** 6
**Confidence:** 3

**Summary:**

The paper introduces Knowledge as Functional representation (KasF), a new method for knowledge reasoning in NLP that models knowledge as dynamic semantic fields. This approach outperforms traditional models on several NLP tasks by requiring fewer parameters and providing more precise knowledge encoding. The main contribution is the innovative functional representation that achieves state-of-the-art results on benchmarks such as CommonsenseQA, with the potential for more efficient and sustainable NLP models.

**Strengths:**

Innovative Knowledge Representation: The KasF model introduces a new functional representation of knowledge that enhances semantic reasoning in NLP.

State-of-the-Art Results: It achieves superior performance on benchmarks such as CommonsenseQA, indicating its potential for accurately handling complex reasoning tasks.

Computational Efficiency: The model's efficiency in terms of parameters used suggests it is less resource-intensive, contributing to more sustainable AI development.

**Weaknesses:**

- Complexity of the method: The KasF model's novel approach might be complex for the broader research community to understand and replicate.

- While the paper shows strong empirical results, it may not thoroughly address how the model generalizes to tasks beyond those evaluated. It's unclear if the approach can be applied effectively to more reasoning tasks that are unseen in training.

- The paper might lack a deeper theoretical discussion on the limitations of the functional representation of knowledge, which would be important for future research to build upon or address its shortcomings.

**Questions:**

- How well does the KasF generalize to unseen datasets that are similar to CommonsenseQA, e.g., SocialIQA, PIQA, RiddleSense?
- How does the KasF model integrate with large external knowledge bases, and what is the impact on its performance when external knowledge is incorporated? Is it possible to do that?
- Is it feasible to connect KasF to the decoder-only models such as Llama?

---

> ### Author Response · Authors · 2023-11-15
> **Point-to-point Response to Reviewer CBXQ (Part I)**
>
> Thank you very much for your review comments. We will address each of your questions and weaknesses, respectively.
>
> Question 1: How well does the KasF generalize to unseen datasets that are similar to CommonsenseQA, e.g., SocialIQA, PIQA, RiddleSense?
>
> Response:
> Thank you for your suggestion. We have expanded our experimentation by utilizing the pre-trained LLaMA2-7B and LLaMA2-13B models on three distinct reasoning benchmarks: CommonsenseQA, SocialiQA, and PiQA. Due to limited computational resources, we opted for a pragmatic approach, replacing the output linear layer (e.g., 4096*32000) with our proposed KasF structure. Subsequently, we fine-tuned the added KasF on each benchmark's training set while keeping the LLaMA's parameters fixed. Our results demonstrate consistent improvements in accuracy on the validation sets.
> |                 | CommonsenseQA | PIQA | Social iQA |
> |-----------------|---------------|------|------------|
> | LLaMA2-7B       | 57.8          | 78.8 | 48.3       |
> | LLaMA2-7B+KasF  | 59.2          | 80.0 | 49.4       |
> | LLaMA2-13B      | 67.3          | 80.5 | 50.3       |
> | LLaMA2-13B+KasF | 69.3          | 81.1 | 51.5       |
>
> It's worth noting that the reasoning ability can potentially be further enhanced, provided richer computational resources and access to external knowledge bases, without imposing constraints on fixing the LLaMA's parameters. We will include the new results into the revision.
>
> Question 2: How does the KasF model integrate with large external knowledge bases, and what is the impact on its performance when external knowledge is incorporated? Is it possible to do that?
>
> Response:
> Certainly, the integration of the KasF module with external knowledge bases is feasible. To elaborate, one can establish a semantic field in the form of a knowledge graph by substituting the dynamic state Q of semantic units with embeddings of entities from a knowledge graph. Following this, we change the metric between Qs as the vector of relations between entities, creating an independent semantic field detached from the input X. In our preliminary experiments conducted on CommonsenseQA (CsQA), we noted that adopting the mentioned approach to convey knowledge related to the query using corresponding triplets in ConceptNet led to an improvement of approximately 3.5% in model accuracy. Nevertheless, due to the evaluation protocol of CsQA that ConceptNet is not allowed to use, we have not included this part of results in the paper.
>
> Furthermore, our findings indicate that by ensuring precision and accuracy in the independent semantic field through manual annotation, the model accuracy could be enhanced to over 94%, showing the significant potential for an enhanced KasF through integration with well-established external knowledge bases.
>
> Based on the above discussion, we will provide more discussion in the revision on the potential of incorporating external knowledge bases.
>
> Question 3: Is it feasible to connect KasF to the decoder-only models such as Llama?
>
> Response:
> First, regarding the integration of KasF with LLaMA for specific reasoning tasks, please refer to the response provided for Question 1.
>
> Second, for the application in generation tasks, we have tried replacing some of the transformer layers in LLaMA with our KasF module, i.e., each KasF module takes text embeddings as the inputs and returns the outputs in terms of Eq 8, which are normalized as the inputs to a subsequent KasF module. The exact building mechanism is like a primary decoder-only LM, except that the transformer module is replaced with a KasF module. More precisely, substituting a standard transformer layer with our KasF module allows us to construct a decoder-only language model from scratch. On a preliminary language modeling task, we observe that constructing a language model using KasF from scratch can yield a competitive Perplexity (PPL) comparable to prominent Language Models (LLMs) like GPT2.
>
> In more detail, the above-mentioned language model, featuring 83.1 million parameters, is trained on the WikiText103 dataset and achieves a test perplexity 19.74. This outperforms other models such as BERT-Large (395 million parameters with a PPL of 20.4), GPT2-large (774 million parameters with a PPL of 22.05), and Hybrid H3 (125 million parameters with a PPL of 23.7).
>
> The above-mentioned results fully prove the feasibility of KasF on decoder-only purposes. Obviously, the answer is YES. We will include more details and information in the revision.

---

> ### Author Response · Authors · 2023-11-15
> **Point-to-point Response to Reviewer CBXQ (Part II)**
>
> Weakness 1 and 2: The KasF model's novel approach might be complex for the broader research community to understand and replicate. How to generalize KasF tasks beyond those evaluated.
>
> Response: Thank you for your valuable suggestion.
> We are committed to improving the accuracy and efficiency of our method to enhance its applicability across a broader spectrum. Notably, our proposed KasF serves as a viable alternative to nonlinear layers, such as the Transformer block, within a language model. As detailed in our response to Question 3, constructing a language model from scratch using KasF yields competitive performance in language modeling tasks when compared to other Large Language Models (LLMs) like GPT2.
> Furthermore, our KasF module can be extended to a knowledge-driven block that explicitly deals with knowledge bases, which support learning with both unstructured data and structured knowledge.
> This suggests the potential of our approach beyond the specific tasks evaluated in the manuscript.
>
> We are confident that, given sufficient computational resources, our method indeed has the capacity to exhibit superior performance across a wider range of reasoning tasks. We will provide more discussion on this point in the revision.
>
>
> Weakness 3: The paper might lack a deeper theoretical discussion on the limitations of the functional representation of knowledge, which would be important for future research to build upon or address its shortcomings.
>
> Response: Thank you for your valuable feedback. In the revision, we plan to incorporate a dedicated section (or in the appendix) that delves into the theoretical underpinnings of our methodology. Regarding the limitations of our approach, they can be categorized into two main points.
>
> 1. CUDA Implementation: One challenge we face is the limited availability of well-established CUDA packages for efficiently computing metric-based operations that are intensively required by our method. Integrating our approach into the Large Language Model (LLM) community demands additional efforts, particularly in terms of CUDA implementation and acceleration.
>
> 2. Hyper-parameter Tuning: The introduction of novel concepts in our method, such as neuronal groups, represents an area that is not extensively explored in the existing literature. Consequently, further investigations are needed to empirically establish the relationships between these concepts and their theoretical implications. We acknowledge the importance of refining the content regarding to these novel elements and will address this concern in the revision by providing additional insights and analyses.

---

> > ### Comment · Reviewer_CBXQ · 2023-12-04
> >
> > Thank you for the feedback. After reviewing both the other reviews and the authors' rebuttal, I've taken into account the overall quality of the paper, and I prefer to maintain my original score.